# Investigating the Impact of Food Rewards on Children’s Motivation to Participate in Sport

**DOI:** 10.3390/children10030432

**Published:** 2023-02-23

**Authors:** Alanna Shwed, Brenda Bruner, Barbi Law, Mark W. Bruner

**Affiliations:** 1School of Health and Exercise Science, The University of British Columbia Okanagan, Kelowna, BC V1V 1V7, Canada; 2School of Physical and Health Education, Nipissing University, North Bay, ON P1B 8L7, Canada

**Keywords:** food rewards, children, sport, motivation, participation

## Abstract

Children who are physically active and involved in organized sport report having the unhealthiest diets. Research suggests excessive calories may be attributed to the prevalence of fast food and candy which are often provided as rewards in sport. This study explored the use of food as a reward in youth sport and the perceived impact it has on children’s motivation to participate in recreational soccer and ice hockey. A multiple instrumental case study approach was utilized. Children aged 4–12 (*n* = 64), parents (*n* = 30), and coaches (*n* = 18) were recruited within central and northeastern Ontario, Canada to participate in focus groups and individual interviews. Transcribed audio recordings underwent inductive thematic analysis. Key themes included: Fun and fast: The culture of food in youth soccer and hockey; (Un)importance of food rewards: The how and why of motivating children in sport; and Youth sport is expensive: Gratitude for sponsorship in youth sport. Themes explain the role of food and food rewards as an element of the youth sport culture as well as the importance of sponsors, regardless of food affiliation, in youth sport. Overall, children’s participation and effort would continue without food rewards; however, they continue to be offered food to motivate and celebrate performance in youth sport. Findings highlight the need to increase knowledge and awareness among parents and coaches on what truly motivates children to help foster healthier strategies for celebrating success and supporting lifelong physical activity.

## 1. Introduction

Extensive research exists to support the physical, psychological, and social health benefits for children engaged in regular physical activity [1,2]. Despite these considerable benefits, children meeting physical activity guidelines also report having the unhealthiest diets [3]. Given the high prevalence of children who are involved in sport to obtain their physical activity (i.e., 60% of children and youth aged 6 to 18 years old) [4], the diets of youth engaged in sport merit important consideration. Understandably, children who are more physically active and involved in organized sports eat larger amounts of food than those who are not [5]. Existing research examining unhealthy diets of children indicates that excess calories are coming from fast food (i.e., food sold from restaurants or snack bars, that are prepared quickly and served in packages ready for take away) [6] and sugar-sweetened beverages [3]. Burgeoning evidence suggests these excessive calories may partly be attributed to candy and sports drinks which are often sold at youth sport centers across North America [7].

In addition to what is available at sport centers, research demonstrates that the choice of snacks is influenced by the highly prevalent fast food sponsorship and celebrity endorsement of unhealthy food (i.e., any food high in saturated fats, trans-fatty acids, free sugars, or salt [8], and do not necessarily come from a restaurant or store) in youth sport [9,10]. Sport celebrity endorsement (e.g., Sidney Crosby, a National Hockey League (NHL) star for Tim Hortons in Canada) and fast food restaurants are major sponsors within youth sport and can influence parents’ and children’s eating habits and preferences [11]. Athletes’ endorsement of these unhealthy products (e.g., Alex Morgan, US Women’s National Soccer Team, and Coca Cola) portrays a false message of health and suggests that unhealthy food is part of the successful sport experience [10]. Further, beyond selling unhealthy food at youth sport centers, there is also heavy marketing within the facilities [12]. Food marketing has led some parents to believe that in order for their child to be successful in sport, they must consume the foods advertised [11]. However, the majority of foods marketed in sport are unhealthy [10] and using food in this manner may play a role in establishing children’s preferences for unhealthy foods [7].

Parents state that despite understanding the negative consequences of feeding their child unhealthy food, they still choose these foods as post-game or practice snacks as they enhance the overall experience and act as a reward for participation [9]. Parents’ rationale for using food rewards to influence children’s behavior might be because it is effective [13]. However, they may also simultaneously offer mixed messages to children about their behavior and what role food should play in their lives [13]. The majority of existing literature exploring how food rewards play a role in parenting has focused on parents using preferred foods (i.e., unhealthy) to reward children for eating healthy foods [14]. Research looking beyond utilizing unhealthy foods to encourage children to eat healthy food is limited and brings into question the short-term and long-term implications of incorporating food rewards as motivation for other behaviors. The use of food rewards for physical activity, and in particular sport participation, is an aspect that is often overlooked; specifically, research investigating food in youth sport and fast food sponsorship as they relate to rewards and motivation for sport continuation is lacking [15]. Investigating rewards and motivation in youth sport is important because although most children are active through sport, the participation rate has dropped by 14% over the last decade [5]. Understanding children’s motivation to participate in sport is critical for supporting long-term engagement and lifelong physical activity [16].

Therefore, the purpose of this research was to explore the use of food as a reward in youth sport and the perceived impact it has on the children’s motivation to participate in recreational soccer and ice hockey (referred to from here on in as hockey). Soccer and hockey are the top two sports for participation among Canadian children [17]. Further, they are the only two sports, out of the top five for participation, that are partnered with unhealthy food companies for sponsorship (i.e., Tim Hortons, PepsiCo, Powerade) [18,19]. The aims of this study were to:Explore the use of food as a reward from the child, parent, and coach perspectives;Explore the use of unhealthy food as a mechanism of motivation for participation;Explore the influence of fast food sponsorship on motivational methods and participation in youth sport.

## 2. Materials and Methods

A constructivist paradigm, with a relativist ontology, and subjective and transactional epistemology [20] guided this multiple instrumental case study [21]. A multiple instrumental case study allowed for the exploration of youth recreation soccer and hockey within the larger idea of the impact of food as a reward in youth sport [21]. Within the context of this study, the larger idea was food as a reward in youth sport and the cases explored was the impact of using food as a reward in youth recreational soccer and hockey. This multiple instrumental case study explored the role between food rewards and unhealthy (i.e., energy-dense, nutrient-poor) food sponsorship on children’s motivation to participate in recreational soccer and hockey. We investigated the influence that food sponsorship in sport has on children’s desire to participate in soccer and hockey, and parents and coaches’ motivational strategies for children’s participation.

Following approval by the institutional research ethics board (Nipissing University Research Ethics Board #: 102217), criterion-based and snowball sampling methods were used to recruit children, parents, and coaches through two youth recreational soccer leagues in northeastern Ontario, Canada (July 2019–August 2019), and two minor hockey associations in central and northeastern Ontario (September 2019–February 2020). Recruitment ended because continued recruitment efforts did not yield more participants. A total of 11 soccer and 21 hockey associations were contacted through email. Ten associations (four soccer, six hockey) responded with interest and shared the study information with coaches and parents. Two associations from each sport had interested participants who emailed the lead author.

After a pilot study to ensure the relevance and appropriateness of the research questions, consent and/or assent were obtained from all participants and focus groups, and interviews were conducted. Focus groups with children aged 4–7 and 8–12 were conducted before or after an already scheduled soccer practice or game (at a booth on the side of the field), or hockey practice (in the hockey arena board room). The age group of children was chosen to understand motivational factors for sport participation across various ages and to capture a time when children first enter sport through to adolescence when participation often drops off [22]. Following a modified graphic elicitation procedure outlined by Cammisa and colleagues [23], children aged 4–7 took part in drawing during their focus groups. Participants were asked to draw themselves playing soccer or hockey, their favorite thing about playing, and what they eat before and after soccer or hockey (drawings can be found in Appendix A). Combining drawing with asking children to verbally answer questions helps reduce their reliance on their own language skills and can give more detailed and organized responses [23]. All child focus groups were conducted separately from parents and coaches to limit their influence [24]. Parent focus groups took place in the same location as the child focus groups; however, they occurred during their child’s practice or game. Coach interviews happened in-person or over the phone at a time most convenient for the individual. Focus groups and interviews followed semi-structured interview guides (see Table 1 for questions) and ended after all questions were asked and participants had nothing to add. Interview questions were developed from the literature [22,25], in consensus with the authorship team, and then piloted with four participants. As a greater understanding of context was obtained, additional questions were added to the interview guides after initial interviews and focus groups.

Each participant was also asked to complete a short survey to better understand their sport background and contextualize the role sport plays in their lives. Surveys were completed either in-person (parents, children, coaches) or emailed to the participant (coaches) to complete, prior to their interview or focus group. Example questions include: How important is sport? and, How many hours a week are you involved in sport?

Descriptive statistics (means, standard deviations) were used to describe the sample population in terms of coach experience, parental time commitment in youth sport, and children’s sport involvement. The focus group and interview audio recordings were transcribed verbatim, and the data were managed using NVivo 12 [26]. Inductive thematic analysis [27] began after the first focus group and ensured we maintained ontological and epistemological authenticity [20]. First, the lead author AS became familiar with the data through conducting the interviews, transcribing the recordings, and reading through the transcripts. AS wrote down initial thoughts but did not start generating codes until the second read through of all transcripts. Codes were generated by grouping meaningful (i.e., relevant to the topic of interest) repeated ideas, and phrases. After all transcripts were coded, broader themes were created by grouping codes and considering how each could be combined to form an overarching theme (codes and themes can be found in Appendix A). BB acted as a critical friend by going through all the codes and challenging the groups of themes until each code group made sense and each theme properly encompassed all codes. AS then presented all themes to the entire authorship team to further refine themes to ensure they aided in telling a useful story in relation to the research question.

To ensure rigor and methodological coherence, criteria explained by Tracy [28] for trustworthiness and Smith and McGannon’s [29] updated recommendations were implemented. To enhance rigor, trustworthiness, sincerity, and credibility [28,30], this study included: (1) multiple participant groups (i.e., children, parents, coaches), methods of data collection (i.e., interviews, focus groups, graphic elicitation), and researcher perspectives (i.e., moderators, critical friend, principal and co-investigators); (2) self-reflexivity practices (e.g., field notes, critical conversations between authors) throughout the entire study; (3) an audit trail of researcher and participants’ insights with a flexible interview guide (i.e., modifications and additions to questions were made as more information was learned throughout data collection); (4) member reflections (i.e., asking for participants to reflect on what the researchers heard); (5) the use of a critical friend (AS, BB); and (6) meetings with the entire research team to determine how to present codes and themes in a way that best reflected the knowledge.

## 3. Results

A total of 112 participants took part in the study (Table 2). Participants comprised 64 children (13 girls, 51 boys; 13 children aged 4–7, 51 children aged 8–12), 30 parents (17 women, 13 men), and 20 coaches (4 women, 14 men). Focus groups (*N* = 20) ranged in size from two to 16 people (*n* = 64 children; *n* = 30 parents). Individual interviews (*n* = 18) were conducted with coaches.

Survey information helped to conceptualize the amount of time participants dedicated to youth sport and how important sport was to them. The experience among the coach participants (*n* = 18) ranged from 1 to 30 years; however, all coaches that took part in the study viewed sport as extremely important. Parents (*n* = 30) involved in the study spent two or more hours a week dedicated to their child’s sport participation and all of them viewed sport as either somewhat or extremely important. Lastly, 70% of children involved in our study said that soccer/hockey is their favorite sport and 92% indicated that sport is either somewhat or extremely important.

Three major themes were constructed from the thematic analysis: Fun and fast: The culture of food in youth soccer and hockey;(Un)importance of food rewards: the how and why of motivating children in sport; andYouth sport is expensive: Gratitude for sponsorship in youth sport.

### 3.1. Fun and Fast: The Culture of Food in Youth Soccer and Hockey

This theme explains why fast food and unhealthy food are both elements of youth soccer and hockey and how the expectation of these foods can play a role in developing children’s motivation. Participants discussed the expectation of food at sport venues and fun team snacks as well as the convenience of fast food.

Many participants discussed what children eat after soccer or hockey at the sport centers. Coaches discussed the types of food that can be found at sport venues: “We play one night…down at [name of field] and they’ve set up the big ice cream truck right at the field…there’s a lot of postgame visits there” (SC6); however, they did not approve: “The greatest thing that ever happened is the [hockey arena’s] canteen is closed for the season so there’s no french fries, pogos [corndogs], you know the fatty fried food” (HC1). Hockey parents explained that depending on the specific hockey arena, there is the potential for healthier food to be offered: “Other rinks [hockey arenas] have great ones…like chicken noodle soup and chili, more healthy stuff. The ones…out of town have gotten better and better every year I’ve noticed, good food” (HFG2 P4). However, children have come to expect unhealthy food at every hockey arena: “They know that there’s going to be treats in here [hockey arena] for sure” (HFG4 P2). The expectation that there is a food stand at every soccer field was not mentioned in this study; however, participants did explain receiving food after soccer is common.

Soccer participants explained team snacks are part of the experience. One parent coach explained that post-game or practice snacks are common but vary by team and family:

“The coach last year for one of my daughter’s teams handed out huge candy things at the last game. Some people are really big on the sport drinks, so I see Gatorades getting passed around. It really varies…if your family does healthy eating, that family is going to bring a healthy snack and other families if that’s not a concern, they’ll bring sugary treats.”(SC6)

Similarly, parents supported what coaches said in that: “The end of the game it’s varied from parents handing out like almost a bag of chips, a chocolate bar, to a freezie [popsicle] or something” (SFG2 P5). Children echoed parents and coaches by saying “Every time a different parent brings a different treat” (SFG2 A3). Although team snacks appeared to be a prominent element of the youth soccer experience, they were not mentioned by hockey participants. However, hockey participants did explain that unhealthy food before or after hockey is part of the experience.

Fast food restaurants were given as examples of where families go to eat before or after hockey; however, the choice of those restaurants was rationalized by the need to eat, and not necessarily their preference for that food. Parents explained they oftentimes need coffee for themselves and use the opportunity to fuel their child before or after hockey:

“My son would have his bagel and his hot chocolate, then my daughter would have her bagel…and then I would have my coffee. Last year that’s what it was when we had early morning practices. It was Tim Hortons, then the game or practice.”(HFG4 P4)

Children also explained that hockey is often around a mealtime and it made sense to eat on the way to the hockey arena and grab something for after: “I’ll wake up in the morning…for an early game you get something from Tim’s [Tim Hortons], usually like a bagel and then I get Timbits [donut holes] or something for after the game” (HFG5 A1). Many discussions from parents and coaches also explained that stopping at a fast food restaurant is often necessary: 

“You do what you have to do as a parent and as a family. It’s not ideal to eat fast food…The other side of the problem is that it would be difficult for practices to be any other time. Hockey doesn’t have a higher priority over school or sleep and there aren’t enough rinks [hockey arenas] to facilitate fitting every team into an ideal timeslot. So, I’d rather eat poorly once a week to have my child learn the rewards of hockey.”(HC8)

Hockey participation is important to families and often dictated dietary habits; however, the same reliance on fast food because of time was not mentioned by soccer participants in this study.

### 3.2. (Un)importance of Food Rewards: The How and Why of Motivating Children in Sport

This theme discusses rewards and their influence in youth recreational soccer and hockey. Participants explained the prominence of food rewards, why they exist, and why they are not the main motivating factor behind children’s sport participation.

Participants indicated that food is often given as a reward. Hockey coaches explained that their most valuable player award often consists of unhealthy food: “I had given them a little certificate to say that they won the player of the game on such date and my, one of my coaches, my assistant coaches, owns a gas bar in town and he’s donated chocolate bars” (HC8). Soccer coaches also said giving out a gift card award to a restaurant chain is required: “In house league [recreational] soccer, I’m expected to hand out a player of the game card…which is sponsored [by] East Side Mario’s [chain restaurant]” (SC6). Children indicated their parents will reward them with food for playing well: “My mom tells me if I get more than five goals then I get a treat” (SFG4 A1).

Some coaches reported using rewards to acknowledge their athletes for displays of effort or success: “It would be the most dedicated, most sportsmanlike, top scorer, those types of things” (HC12). Parents said coaches also reward effort in practice: “Last year [coach] gave a Gatorade to the best listener every practice” (HFG4 P4). Parents also reward children, and coaches indicated overhearing parents explaining to their children how they can earn a treat: “We have a couple of kids on the team the parents will go “okay if you do this well you get a Gatorade, if you don’t play well, you don’t get one” kind of thing” (HC13). Hockey parents did not explicitly admit they give their children rewards for participation; however, they did explain that when hockey is early in the morning or if practice is more skills-focused (e.g., power-skating) and not a game, it is more difficult to motivate their child to get ready to go to hockey. One parent said Tim Hortons helps his son get up for early hockey: “Early mornings I occasionally have to throw in a Tim Hortons visit or something” (HFG1 P4). In contrast to hockey parents, soccer parents did indicate that they reward their children for participation: “So, if you’re trying your hardest, you’re running hard, and you’re listening to your coach and you listen to the advice I give you before the game, you’re going to get rewarded” (SFG2 P4). They rationalized the reward because of the hard work displayed by the child: “It’s like here’s your Gatorade because you just ran for two hours and sweat your ass off right” (SFG2 P2). Children supported coaches and parents by explaining how they earn rewards: “Good games, if I played well…skated well, made some good plays, scored some goals” (HFG4 A1) and “Just try our try our best” (SFG4 A1).

Despite almost all coach and parent participants indicating that they reward children, they also said that those rewards ultimately do not change effort or participation: “I would say they are excited by them [treats] and enjoy them, but it’s certainly not why they’re there…The treats are a perk, but I don’t think that’s what gets her to the field” (SC6). However, coaches did explain there are positives that come from rewarding children:

“I don’t think at the end of the day…the candy in the lobby is necessarily enough for a kid to not get involved. But I do think that there are positive experiences that are coming from those things that a child remembers.”(HC11)

When asked about how important they think the snacks and rewards are to their children and the reason they play, every parent said they were not important as: “It’s like a bonus, it’s not a motivator…They want to play hockey, they want to score goals, you don’t have to motivate them, they want to go out there and score a goal” (HFG1 P1). Children supported what coaches and parents said by indicating they do not rely on the rewards to participate. When asked if they would still play if they were not given rewards every child said: “I would still play” (HFG7 A13). However, children also appreciated the rewards. Children in soccer explained the post-game rewards help ease the pain of a tough loss: “But on those…days when I did something that I think I did bad[ly] like scored in [my] own goal…then those rewards are really needed to boost my confidence because… I don’t even know if I’d be here [playing soccer] today” (SFG2 A1).

When asking parents about why rewards and snacks an element of youth sport despite them not being critical for ensuring continued participation, hockey participants suggested they are a part of the tradition: “I believe the treats are more about tradition” (HC9). One parent explained the only reason these rewards are now an expectation is because they were introduced:

“I think had my uncle not done it at the beginning, I don’t think it would have ever become a thing because he’s [child] not used to getting junk [unhealthy food] and stuff like that. But now…it’s routine…to him [child] hockey equals McDonald’s.”(HFG5 P1)

Parents and coaches suggested the food at games and practices are something children look forward to as part of the tradition; however, they also enjoy the sport for just the sport itself.

Exploring why the absence of rewards would not influence participation rates led to parents explaining they did not need to motivate their child to play. A soccer parent said: “Yeah if I were to tell [name of child] that I wasn’t going to take him to soccer he would throw a fit” (SFG1 P1) and hockey parents similarly said: “I don’t need to motivate at all, my kids want to be here all the time” (HFG4 P1). Children supported what their parents said by suggesting they value the sport more than the rewards they receive from parents and coaches: “I don’t care about the candy; I care about the game” (HFG2 A3) and “I don’t need anything to encourage me to go” (SFG4 A4). It was obvious the children involved in this study truly enjoy playing soccer or hockey without the food rewards.

### 3.3. Youth Sport Is Expensive: Gratitude for Sponsorship in Youth Sport

This theme discusses role of sponsorship on participation in youth sport. Participants did not indicate that sponsors influenced their motivation; however, they valued the financial support that help enable participation.

When asked about their opinion, most participants (parents, coaches, and children) initially responded by saying they did not have an opinion. When probed further or after considering the question for longer, many participants expressed their gratitude for the support youth soccer and hockey receive from sponsorship. One hockey coach explained, “You know I think…if I had an opinion of them, I’d say thank you very much. If it wasn’t for sponsorship in sport, where would we get the money to run associations” (HC6). Parents in both soccer and hockey talked about how sponsorship helps reduce the cost of their child playing: “If it cuts down the cost for us great…It [sport] gets to be pretty expensive ‘cause once you start putting more than one kid through it” (SFG1 P2).

Despite participants’ appreciation for the financial support, parents and coaches explained that there are standards for what companies are allowed to sponsor youth sports: “Bars/pubs…strip joints…E-cigarette stores are not appropriate. You want local businesses to chip in so that people see that they are investing back in the community” (HC12). Participants did not discuss sponsors related to the food children eat; however, coaches said financial support is more important than ensuring all sponsors are healthy food companies: 

“I understand why some individuals may not want to see a fast-food chain advertised but…without their sponsorship, league/team fees would increase which may lead to some children not having the opportunity to play hockey as it is expensive.”(HC12)

Coaches emphasized the fact that, without the help of sponsors, regardless of what food the company represents, youth sport would not be able to run: “We need money so desperately that so long as it’s coming from an ethical source…we’re not turning it away because it’s McDonald’s” (SC3).

## 4. Discussion

This study aimed to examine the use of food as a reward in youth recreational soccer and hockey from child, parent, and coach perspectives. Unhealthy food rewards were prevalent, supporting previous literature examining the food environment and food rewards in youth sport [15,31]. Research has found that parents often use unhealthy food such as candy or chips for post-game treats [9,15], and they are common in youth recreational leagues [32]. However, participants in this study suggested that unhealthy food in soccer and hockey may not be solely used for providing rewards.

Hockey parents explained that feeding children with fast food is necessary because of time constraints before or between games, which are normally scheduled around mealtimes. The theme of limited time dictating food choices supports previous literature that child participation in recreational sport leads parents to make changes to their family schedule and structure [31]. Further, although fast food is purchased because of time constraints, this reliance on convenience foods is arguably contributing to the association children have with hockey and receiving an unhealthy food reward. Interestingly, the theme of limited time dictating food choices was not discussed among soccer participants, which might be explained by the difference in timing of the sports. The recreational soccer season in Canada predominantly runs in the summer months (May–August) when children are not in school and work hours of parents might be more flexible. On the other hand, hockey runs during the fall and winter months (September–March) during school, with normal work schedules for many parents. Lastly, although socioeconomic status information was not collected, hockey is an expensive sport, even at the recreational level, and Tim Hortons and McDonalds are inexpensive; for some families there may be economic reasons behind their choice of food to feed their child. Future research is warranted to investigate the economic influence of youth sport on family eating habits.

While soccer parents, coaches, and children did not suggest that fast food is an element of youth soccer, all participants in both soccer and hockey did indicate that food rewards are common and an expected part of the sport. Other research has found similar themes from children explaining they expect certain types of food (e.g., French fries) as rewards after hockey because they are part of the routine [33]. However, our study also found that the choice of food rewards is heavily dependent on family habits and what each one deems appropriate. For example, post-game food rewards ranged from slushies from the concession stand to home-prepared sliced watermelon.

The contrast in beliefs of what was considered a food reward reflects similar findings to Rafferty and colleagues [9] who found perceptions of team snacks varied between age groups. Parents of younger children expressed concern for ensuring snacks are healthy, whereas parents of older children were more accepting of the typically offered snacks in sport (i.e., unhealthy foods) [9]. No noticeable differences in post-game food rewards between the different ages of children emerged in our study; however, there was a very clear difference in family habits among and within individual teams. Regardless of the contrast in what was considered a treat, in this study parents and coaches explained that food rewards are not common outside of sport for their children and are something normally just associated with soccer or hockey. Children supported their parents by saying they do not always receive treats outside of soccer or hockey but indicated that food rewards from sport happen often (i.e., potentially twice a week for a game and practice).

The second aim of this study was to understand the use of unhealthy food as a mechanism of motivation for sport participation among children and youth. Although there is substantial literature on motivation in youth sport, studies exploring the use of food as a reward and its potential influence on motivation is scarce. Most studies have focused on adolescents, and none have included children, particularly the youngest entering sport (i.e., age 4–5). In addition, few studies have explored the role of food in the sport beyond health-related outcomes (e.g., obesity) and dietary habits. Therefore, the findings from this study add to understanding the role of food rewards on children’s motivation to participate in sport. Parents and coaches reported that rewards such as food sometimes help promote effort from children. This supports the previous literature around parents’ use of rewards to evoke a desired behavior from their children [14], and the reasons children are given food rewards after sports [15,31]. Despite parents and coaches indicating that food rewards can help promote effort, motivation, and fun, they are not what children value most or what influences their involvement in soccer or hockey.

This study reveals an interesting paradox where parents, coaches, and children all indicated that food rewards do not influence motivation; however, they continue to be utilized. We found that parents and coaches provide food rewards to make the experience positive, celebrate success, continue tradition, and encourage effort, but not to motivate participation. Although children would play soccer or hockey without the treats, the food rewards give them something to look forward to, and they viewed playing soccer or hockey and getting rewarded as a “win, win”. While rewards were not found to be important for participation, they do add to the enjoyment of sport, which ultimately leads to sport continuation [33]. Similarly, research by Elliott and colleagues [15] also found that receiving food rewards were seen as a positive element to being involved in sport by children. Despite the use of rewards in soccer and hockey, parents, coaches, and children emphasized that participation would not look different if these rewards were not a part of soccer and hockey.

The finding of parents and coaches using rewards to promote and celebrate effort and success, but ultimately not aiding in children’s motivation to participate or try their best, may be explained by the type of children participating in this study. Children who like soccer or hockey may not be reliant on food rewards to fuel their motivation to participate or put their best effort forward. However, children in this study did appreciate the food rewards, which might be because they value the meaning behind the reward (i.e., acknowledgement, praise for their hard work). Further, it is clear from these results that what motivates children to play most are the successes found in the game (e.g., scoring) and playing with teammates. Children wanting to play soccer and hockey because of the natural outcomes that come with participation and not needing to be motivated by food rewards suggests greater intrinsic motivation [22]. Future research should continue to explore and better understand why parents and coaches continue to reward children for behavior they maintain will already take place or continue without the reinforcement. Further, exploring parents and coaches’ motives behind the use of food rewards might aid in developing education for parents and coaches that resonates with them to help implement effective and healthy strategies for motivating children.

Lastly, this study explored the influence of fast food sponsorship on motivation and sport participation among children. All participants could list many different sponsors for youth soccer and hockey; however, there was no conscious influence on choice of treat or motivational methods found in this study. Most participants did not have an opinion on the type of sponsorship except for their gratitude for financial support, which is similar to the findings of Kelly and colleagues [34,35,36]. Unlike previous literature though, the current study did not find that participants had a problem with fast food sponsorship. Most children did not care who sponsored them as long as they got to play, and parents and coaches were only concerned about the appropriateness of the sponsor in the youth context. While parents, coaches, and children in either sport did not frequently consider the potential influence of sponsors, most hockey participants did recognize Tim Hortons as a well-known sponsor in hockey. As Tim Hortons is a Canadian brand which uses high profile NHL players such as Sidney Crosby, Wayne Gretzky, and Tim Horton himself, it is unknown if this is specific to the Canadian culture, or if the association of a fast food restaurant with a particular sport exists elsewhere in a different context. It might be that the influence of fast food and celebrity endorsement (i.e., Sidney Crosby for Tim Hortons) does have an implicit effect on behavior; however, it is at an unconscious level and so is not recognized or acknowledged upon reflection. Research that does not rely on methods of reflection is warranted to capture the influence of fast food sponsorship and food availability on sport participation behavior in the youth sport context.

There are several strengths of this research that are important to highlight. This study supports previous literature from the American [9,31] and Australian youth sport setting [15,34,35,36], and has begun to explore food rewards in youth sport in a Canadian context. Research can build on these cases to look beyond youth recreational soccer and hockey in central and northeastern Ontario and children who are already intrinsically motivated. Second, this study examined the perspectives of parents, children, and coaches. Exploring individual realities among people in the same group highlighted the influence of various levels and factors on behavior. Third, this study included children who are just starting sport. Most research examining children’s motivation in sport has looked at children over the age of 11, even though children enter sport much younger [37]. Lastly, this study was a qualitative exploration, allowing for direct contact with participants and discussion where the researcher was not the expert [38]. Participants’ opinions and perspectives were kept intact by allowing them to share their own thoughts instead of categorizing and rating their experiences through predetermined quantitative measures [38].

While this study adds to the literature exploring physical activity and motivation in youth sport, there are some acknowledged limitations. First, there was the potential for social desirability to emerge during focus groups [39]. To help limit social desirability tendencies (e.g., vague answers, inconsistent responses), researchers provided reassurance to participants that all opinions were valued, probes for more information were used, and requests for examples were made to accompany responses. As well, notes were taken throughout the focus group to consider if answers given by participants truly reflected their honest opinions or if social desirability was perceived. Finally, it is possible those who chose not to participate are the ones who rely on food rewards as motivation. All children in this study liked soccer and hockey and genuinely looked forward to playing. It is possible that children who find sport to be important are motivated differently than children who do not enjoy the sport. Therefore, there is a need to understand what methods of motivation work for children who are not intrinsically motivated to play various sports to encourage sport continuation [40] and foster lifelong physical activity habits [16].

## 5. Conclusions

The findings of this research indicate that food rewards are common occurrences in youth soccer and hockey for effort, success, and participation in both games and practices. However, despite food rewards being considered a normal aspect of the sport experience, they do not appear to influence motivation to participate. While children enjoy receiving treats, they are not what children value most. Knowing that using food as a reward is not required to motivate children’s participation in our sample of youth soccer and hockey players suggests a need to promote other methods of motivation. Limiting food rewards and implementing healthier foods or alternative non-food rewards are recommended to continue to help foster enjoyment and sport continuation but also to help minimize the association that has emerged between youth sport and unhealthy food. Further, providing coaches and parents with information about healthy methods for motivating children and strategies that are not unhealthy food rewards, might help facilitate and foster the more intrinsic reasons children enjoy participating. 

## Figures and Tables

**Table 1 children-10-00432-t001:** Semi-Structured Interview Guides.

Group	Guiding Questions
Children (aged 4–7)	Why did you start playing soccer/hockey? What do you get most excited about for soccer/hockey? Do you have any routines (things you normally do) before and/or after your soccer/hockey practice or games? What are the kinds of things you eat before and/or after your soccer/hockey practice and game? Do you ever get a treat/reward at soccer/hockey? What kind do you get? How often do you ever not want to go to soccer/hockey or try your best? What makes you want to go anyways? What do you know about [team sponsor name]? How do you feel about them?
Children (aged 8–12)	Why did you start playing soccer/hockey? What is your favorite part about playing soccer/hockey? What types of foods do you eat before and/or after playing soccer/hockey? Explain to me the differences between what you eat when it’s a practice day versus a game day, if there is a difference, and why you eat those foods. Do parents bring or offer snacks? What kinds of snacks do they bring, and which ones do you prefer? Tell me a little about what it’s like to get those snacks. What do you know about [team sponsor name]? How do you feel about them?
Parents	How did your child get involved in soccer/hockey? What is your child’s favorite part of playing soccer/hockey? How do you motivate your child to go to soccer/hockey? If you child is misbehaving or disengaging from the practice or game, what strategies do you use to get their attention and focus back on track? What kinds of food does your child eat before and/or after soccer/hockey? Does what they eat depend on if it is a practice versus a game? Does anything special happen if your child scores a goal or plays well? Do you have any routines before and/or after soccer/hockey? Does your child ever receive post-game snacks or treats and how do they earn them? What are their favorite ones to get? Is there anything that is given out to the kids from the organization throughout the season? What do you think of [sponsors name]? What are your opinions on the types of sponsors in youth sport? What do you consider an appropriate source of sponsorship for youth sport and why? We have heard that treats and awards are often used to reward effort and success, but parents, coaches, and children indicate that without them, participation and effort would remain the same. What are your thoughts on using treats, knowing that kids say they would play without them? * We heard that the timing of games and practices often influence meal choice. What are your thoughts on having to choose between making quality meals versus quick, convenient meals due to limited time? *
Coaches	How did you get involved in coaching? How do you keep children motivated on your team? How do parents help motivate their children? Is there anything that is given out to the kids from the organization throughout the season? Does your team have snacks after practice or games? Who provides the snacks? What do the snacks normally consist of? How often do kids ask for snacks and what do they prefer to receive? What do you think of [sponsors name]? What is your opinion on the types of sponsors in youth sport? What do you consider an appropriate source of sponsorship for youth sport and why? We have heard that treats and awards are often used to reward effort and success, but parents, coaches, and children indicate that without them, participation and effort would remain the same. What are your thoughts on using treats, knowing that kids say they would play without them? * We heard that the timing of games and practices often influence meal choice. What are your thoughts on having to choose between making quality meals versus quick, convenient meals due to limited time? *

Note. * Questions added after initial interviews and focus groups.

**Table 2 children-10-00432-t002:** Participant Sport and Demographic Information.

	Soccer		Hockey	
	Mean (SD)	Frequency (%)	Mean (SD)	Frequency (%)
**Coaches**				
Years of coaching experience	13.2 (7.81)	-	8.4 (8.65)	-
Number of youth sport teams currently coaching	1 (0)	-	1.6 (0.79)	-
Number of Sports Involved in as a Child	4.5 (2.88)	-	5.7 (2.19)	-
Time Commitment to Coaching				
Less than 1 hour/week	-	0 (0)	-	1 (8)
Between 2 and 3 hours/week	-	1 (16.5)	-	4 (33)
Between 4 and 9 hours/week	-	4 (67)	-	6 (50)
10+ hours/week	-	1 (16.5)	-	1 (8)
Importance of Sport				
Extremely Important	-	6 (100)	-	12 (100)
Somewhat Important	-	0 (0)	-	0 (0)
Somewhat Unimportant	-	0 (0)	-	0 (0)
Not at all Important	-	0 (0)	-	0 (0)
**Parents**				
Number of Sports Involved in as a Child	2.7 (1.11)	-	3.0 (2.1)	-
Time Commitment to Child’s Sport				
Less than 1 hour/week	-	0 (0)	-	0 (0)
Between 2 and 3 hours/week	-	1 (14)	-	10 (42)
Between 4 and 9 hours/week	-	3 (43)	-	12 (50)
10+ hours/week	-	3 (43)	-	2 (8)
Importance of Sport				
Extremely Important	-	5 (71)	-	13 (54)
Somewhat Important	-	2 (29)	-	11 (46)
Somewhat Unimportant	-	0 (0)	-	0 (0)
Not at all Important	-	0 (0)	-	0 (0)
**Children**				
Number of Current Sports	2.1 (1.06)	-	2.0 (1.27)	-
Favorite Sport is Soccer/Hockey	-	13 (54)	-	32 (82)
Importance of Sport				
Extremely Important	-	17 (71)	-	24 (62)
Somewhat Important	-	7 (29)	-	11 (28)
Somewhat Unimportant	-	0 (0)	-	2 (5)
Not at all Important	-	0 (0)	-	2 (5)

## Data Availability

Data is contained within the article or Appendix A.

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
