# Peer review of "Investigating the Impact of Food Rewards on Children’s Motivation to Participate in Sport"

_children, 2023, doi:10.3390/children10030432_

Round 1
Reviewer 1 Report
The authors raise a very important issue regarding „Investigating the impact of food rewards on children's motivation to participate in sport”.
This problem is very current and has a global reach. Unfortunately, despite the progress of research in this field, the results of physical activity and proper nutrition among children and adolescents are still not satisfactory. The results of the presented studies confirm this fact, but their advantage is the indication of the connection between physical activity and the way of eating of children and adolescents.
The article should be treated as an important contribution to the general campaign to promote physical activity and proper nutrition of children and youth, together with social support.
Author Response
Thank you so much for taking the time to review our manuscript.
Reviewer 2 Report
Title: Investigating the impact of food rewards on children’s motivation to participate in sport
Summary
In this paper, the authors have examined the use of food as a reward in youth sport and its influence on children’s motivation to participate in recreational soccer and ice hockey. Participants were 64 children, 30 parents as well as 18 coaches who participated in this study. Transcribed audio recordings underwent inductive thematic analysis. The authors highlighted the need to increase knowledge and awareness among parents and coaches on what truly motivates children to help foster healthier strategies for celebrating success and supporting lifelong physical activity.
Evaluation
Thank you for giving me the chance to review this manuscript. In my opinion, the topic of this study is interesting for publication in the Journal. In addition, the design for the study is appropriate to answer the research questions, and the paper is well written. For a paper, the manuscript is quite straightforward and so I think that can be acceptable for publication. However, some points and suggestions should be addressed by the authors, in order to improve the quality of the manuscript.
Minor points and suggestions
Please add some information about the participants such as their mean age, gender, …
In the current form, the abstract is a little vague, especially the results section, please rewrite it in a better form.
The introduction is written in a good manner, however, please add research hypothesis to the end of the introduction.
How did you calculate the sample size?
About questions regarding the interview, how are questions made? On what basis have these questions been raised?
In table 3, for “Number of youth sport teams currently coaching” and “Number of Sports Involved in as a Child”, the authors write “mean”. It is not common for numbers (number of coaches, number of sports). Write the mean. It is better to write only the frequencies.
